# PtOEP–PDMS-Based Optical Oxygen Sensor

**DOI:** 10.3390/s21165645

**Published:** 2021-08-21

**Authors:** Camila M. Penso, João L. Rocha, Marcos S. Martins, Paulo J. Sousa, Vânia C. Pinto, Graça Minas, Maria M. Silva, Luís M. Goncalves

**Affiliations:** 1Center for MicroElectromechanical Systems (CMEMS-UMinho), University of Minho, Azurém, 4800-058 Guimarães, Portugal; a78906@alunos.uminho.pt (C.M.P.); b8874@dei.uminho.pt (J.L.R.); mmartins@dei.uminho.pt (M.S.M.); psousa@dei.uminho.pt (P.J.S.); vpinto@dei.uminho.pt (V.C.P.); lgoncalves@dei.uminho.pt (L.M.G.); 2Chemistry Department and Center of Chemistry, University of Minho, Gualtar, 4710-057 Braga, Portugal; nini@quimica.uminho.pt

**Keywords:** optical sensor, PtOEP, PDMS, oceanography

## Abstract

The advanced and widespread use of microfluidic devices, which are usually fabricated in polydimethylsiloxane (PDMS), requires the integration of many sensors, always compatible with microfluidic fabrication processes. Moreover, current limitations of the existing optical and electrochemical oxygen sensors regarding long-term stability due to sensor degradation, biofouling, fabrication processes and cost have led to the development of new approaches. Thus, this manuscript reports the development, fabrication and characterization of a low-cost and highly sensitive dissolved oxygen optical sensor based on a membrane of PDMS doped with platinum octaethylporphyrin (PtOEP) film, fabricated using standard microfluidic materials and processes. The excellent mechanical and chemical properties (high permeability to oxygen, anti-biofouling characteristics) of PDMS result in membranes with superior sensitivity compared with other matrix materials. The wide use of PtOEP in sensing applications, due to its advantage of being easily synthesized using microtechnologies, its strong phosphorescence at room temperature with a quantum yield close to 50%, its excellent Strokes Shift as well as its relatively long lifetime (75 µs), provide the suitable conditions for the development of a miniaturized luminescence optical oxygen sensor allowing long-term applications. The influence of the PDMS film thickness (0.1–2.5 mm) and the PtOEP concentration (363, 545, 727 ppm) in luminescent properties are presented. This enables to achieve low detection levels in a gas media range from 0.5% up to 20%, and in liquid media from 0.5 mg/L up to 3.3 mg/L at 1 atm, 25 °C. As a result, we propose a simple and cost-effective system based on a LED membrane photodiode system to detect low oxygen concentrations for in situ applications.

## 1. Introduction

The need for oxygen analysis led to the inevitable search for more efficient, robust and low-cost technologies to detect $O_{2}$ in a variety of media and applications. The analysis of DOC (dissolved oxygen concentration) is required in several areas, namely in biology, oceanography, chemistry, the food and pharmaceutical industries, among others. This work focuses on in situ ocean applications, particularly for low oxygen zones. This technology uses standard PDMS fabrication processes and could be potentially useful to analyze oxygen in microfluidic channels, for medical devices, biology and chemistry applications, marine research and other lab-on-a-chip devices.

The most available oxygen sensors are based on electrochemical or optical detection methods. The former, despite providing low-cost and label-free-based sensors, need periodic calibration adjustments and regular maintenance whenever external conditions change. Additionally, they consume oxygen in their reading process, invalidating the results, which require a constant flow of fluid so that oxygen is not completely consumed. Furthermore, they can be affected by the presence of different gaseous species influencing the basic readings. The substances that most influence the readings are hydrogen sulphide, sulphur dioxide, chlorine, carbon monoxide, nitrous oxide, nitric oxide, halogen, hydrogen, ozone and strong substances such as acids and alkaline solutions [1]. In contrast, the optical sensing methodologies do not need periodic calibration, maintaining the reliable readings for a long time, and do not consume oxygen in the reading process, giving independence from the flow [2]. Additionally, one of the biggest advantages of optical sensors for long-term applications is their minimal maintenance. For the ones based on a membrane with an oxygen indicator, the maintenance is practically restricted to the renewal of the luminescent indicator through the annual change of the membrane. However, despite optical oxygen sensors overcoming some of the greatest inconveniences of the electrochemical sensors, they use a recent and innovative principle and they have not yet been able to establish themselves in force in the market [3]. Nevertheless, due to the described advantages of optical sensing methodologies, the herein proposed oxygen sensor is based on an optical approach. Therefore, in the subsequent subsection a background about this sensing methodology is described. In this work we introduce a study about the influence of concentration, of thickness and of solvent on the membrane response. This is paired with a hardware/software developed to optimize cost.

### 1.1. Optical Sensing Principle for Oxygen Quantification 

Optical sensors for oxygen quantification are usually based on the physical phenomenon of luminescence, namely fluorescence and phosphorescence [4]. Fluorescence is obtained by the absorption and excitation of sensitive molecules (present in the indicator) and subsequent transfer of energy, resulting in emission at a higher wavelength. Figure 1 presents a typical optical oxygen sensor, where the membrane is excited with a light source and the luminescence is detected with a photodetector. Optical oxygen sensors are based on the dependence of fluorescence intensity on the concentration of dissolved oxygen. In such sensors, besides the fluorescence of the indicator (doped in the membrane matrix), the membrane needs to be able to diffuse oxygen, in such a way that oxygen will rapidly reach many of the active indicator molecules.

To determine the oxygen concentration in a homogeneous matrix (ideal matrix) where the oxygen accesses all the polymer molecules in an equal way and the indicator has a uniform concentration, Stern–Volmer equations are used, as shown in Equations (1)–(3) [5].
(1)$$\frac{I_{0}}{I}=\frac{\tau_{0}}{\tau }=1+K_{SV}pO_{2}\;$$
(2)$$K_{SV}=k_{q}\tau_{0}$$
(3)$$k_{q}=\alpha D$$
where $I_{0}$ is the maximum intensity of the membrane luminescence (in an oxygen-free environment),  I corresponds to the intensity of the membrane in the sample, $\tau_{0}$ corresponds to the excitation lifetime in a sample without oxygen, τ is the lifetime of excitation of the test sample, $K_{SV}$ corresponds to the Stern–Volmer constant, $pO_{2}$ corresponds to the partial pressure of oxygen in the sample and $k_{q}$ is the quenching response constant. The variables α—solubility constant and D—oxygen diffusion constant represent the properties of the membrane matrix. When it is used an heterogeneous membrane, it is necessary to resort to more complex mathematical equations [6]. As a result of these equations, a linear relation is obtained between the intensity—I (or lifetime—*τ*) and oxygen concentration, describing the distribution of the indicator as homogeneous and uniform; that is, all the indicator molecules are equally accessible throughout the entire membrane.

We present an optical sensor for dissolved oxygen, based on a LED membrane (PtOEP-PDMS) photodiode system to detect low oxygen concentrations for in situ applications. We demonstrate that the signals obtained from the sensor can be used for oxygen measurements in the range from 0.5% up to 20% for gaseous media, and from 0.5 mg/L up to 3.3 mg/L for liquid media at 1 atm, 25 °C. 

### 1.2. PtOEP Indicator

A full review on luminescent indicators is found in [7]. They can be classified as polycyclic, aromatic hydrocarbons, polypyridyl complexes, metal porphyrins, cyclometallated complexes and complexes with rarely used central atoms.

In this work, a metalloporphyrin indicator dye, platinum octaethylporphyrin (PtOEP) is used, due to their strong room-temperature phosphorescence with a quantum yield (φ) of about 0.5 [8]. This indicator has been widely used in optical oxygen sensors and was immobilized in various oxygen-permeable polymeric matrices such as polystyrene [9], poly(aryl etherketone) [10], ethyl cellulose, cellulose acetate butyrate and polyvinylchloride [11] styrene-pentafuorostyrene copolymer film [12], PDMS [13] and poly(1-trimethylsilyl-1-propyne) [14]. A general disadvantage of this indicator is its rather low photostability [7]. However, Platinum tetrakis (pentafuorophenyl)-porphyrin (PtTFPP) can replace PtOEP for higher photostability [9], but its synthesis is more complex and it performs a lower quantum yield (0.21) [15].

### 1.3. PDMS-Based Membranes

To protect the indicator from interactions with the external environment and preserve its integrity and optical characteristics, it is usual, in the literature and in commercially available sensors, to use membranes where indicators are dispersed and immobilized in matrices. However, this membrane should be permeable enough to oxygen, so it can reach the indicator. Depending on the indicator to be used, there are several polymers that can be used to immobilize indicators. The environment (matrix) where the indicator is dispersed has a great influence on the sensitivity of the sensor. Thus, it is common to find indicators that, when immobilized in different matrices, result in highly different sensors, with different sensitivities and ranges.

When choosing the matrix polymer, some factors that should be maximized are the permeability of the matrix to oxygen (*P*), the dispersion (*D*) of the indicator in the polymer homogeneously, the solubility of the oxygen in the polymer (*S*), the compatibility of the indicator solvent with the selected polymer and issues related to the commercial availability of the polymer. The permeability of the matrix to oxygen is given by the following Equation (4).
(4)P≈DS

Polydimethylsiloxane, also called PDMS, has excellent properties as a dispersion polymer, as shown in Table 1, where several dispersion polymers are compared [16,17,18].

PDMS is an interesting polymer that, in addition to completing the chemical and physical requirements discussed above, it is easily synthesized in the laboratory, has great commercial availability at low prices, is non-toxic, is relatively inert and non-flammable, is hydrophobic and wide used in microfluidic systems [19]. PDMS is also impermeable to liquids, thus isolating the sensor electronics.

## 2. Materials and Methods

### 2.1. Membrane Fabrication 

The PDMS solution was prepared considering manufacturer instructions. This solution results from the mixing of 1 g of curing agent (cross-linker) with 10 g of pre-polymer, (Sylgard 184, Ellsworth Adhesives Ibérica, Madrid, Spain) mixed manually with a spatula in a glass beaker. The mixture was vacuumed several times until it became homogeneous, transparent and without air bubbles. Subsequently, 10 mL of the solvent (toluene or THF, Sigma-Aldrich Co. Ltd., Schnelldorf, Germany) was mixed with indicator (PtOEP, Sigma -Aldrich Co. Ltd., Schnelldorf, Germany). A magnet bead was added, the recipient was covered with parafilm and stirred for 15 min. Afterwards, the beaker was uncovered, allowing the solvent to evaporate at room temperature (22 °C, 50%RH) for an additional 15 min in the stirrer. 

The PDMS mixture was added to the indicator, mixed with a spatula and followed by magnetic stirring for 15 min, uncovered. The mixture was covered with perforated parafilm, and ultrasound mixing was carried out for 20 min (maximum effective power) at 26 °C. 

The membranes were fabricated by molding (to obtain thicker films), resulting in thicknesses of 0.1 mm to 2.5 mm. All membranes were placed in the oven for 24 h at 80 °C, to cure PDMS and to remove the solvent. The equipment used for this procedure were: ultrasound Martin Walter Powersonic D, magnetic Stirrer MSH 300, spin-coating Polos SPS Spin 150 and Hot Plate Prazitherm PZ28-2 Heating plate. The thickness of the membranes was measured and non-uniformity in thickness was found (up to 15%) in the membranes area, as a result of the molding process.

### 2.2. Sensor Design and Electronics

The sensor structure was fabricated by additive manufacturing, using ABS (Acrylonitrile Butadiene Styrene) as raw material, with accurate dimensions to support and immobilize the photodiode, filter and LED. The external encapsulation is made of PVC (Polyvinyl chloride). Furthermore, the interior of PVC was painted with conductive paint and connected to the ground of the circuit. Shielded coaxial cables were used to connect both the LED and the photodiode to the electronic circuit. These two strategies will allow noise reduction and protection from external activity (such as movement and capacity changes that interferes with the low photodiode current). Figure 2 represents the sensor section design and Figure 3 a photograph of the fabricated device. 

The sensor includes two electronic circuits: one for signal conditioning (a transimpedance amplifier—TIA) and other for LED excitation (current sink) that included a current sink circuit ensuring a constant current in the LED, thus a constant light intensity. A microcontroller (STM32F767) is used for data acquisition and control. It controls the LED excitation and computes the oxygen concentration, using an internal Analog Digital Converter ADC (12 bits) to convert the voltage output of the signal conditioning circuit. An average of ten readings is realized, and the correspondent oxygen concentrations are calculated and sent by the UART of the microcontroller every 10 s. To reduce energy consumption and to increase the photostability of the membrane, the LED is only turned on to perform the measurement. The hardware is briefly described in Figure 4.

The intensity of the LED light needs to be precise and stable to avoid error in the oxygen reading. For this purpose, a current sink circuit was developed (Figure 5), capable of supplying a constant current independently of the supply voltage fluctuations from the power supply. For the excitation, the UV LED Bivar UV5TZ-385-30 was selected because it has an emission peak at a wavelength (385 nm) close to the maximum absorption of the indicator. 

The voltage in *R*_4_ is proportional to the current of the LED and is compared with a reference voltage (*V*_*REF*_), at the positive input of the amplifier. This comparison result is used to control the gate of the N mosfet (2N7000) and, consequently, the current in the LED in a closed-loop cycle. The LED current is given by Equation (5), whereas *R*_4_ is a precision resistor.
(5)$$I=\frac{V_{REF}}{R_{4}}$$

*C*_1_ and *R*_3_ allow compensating for unwanted oscillations of the amplifier (loop compensation). The value of *R*_3_ must be much higher than *R*_4_, and the *R*_3_*C*_1_ time constant should allow a settling time for the desired switching speed. It was defined at a maximum frequency of approximately 10 kHz, with *R*_3_ of 1.1 kΩ and *C*_1_ of 10 nF. The LED current was defined as the maximum recommended current (20 mA), and a maximum direct voltage of 3.6 V is expected. 

The transistor has a drain-source resistance of less than 10 Ω when turned on, which represents a drain-source voltage below 0.2 V. Using a reference of 400 mV (NCP135BMT040TBG), a supply voltage greater than 4.2 V is necessary, so 5.0 V is used. 

The conversion of the luminescence intensity in a constant voltage is performed using the circuit presented in Figure 6. The selected photodiode (Hamamatsu S1336-8BK) is used in photovoltaic mode, polarized with 0 V, to minimize dark current. This photodiode has a sensitive area of 5.8 × 5.8 mm, low dark current (100 pA) and excellent quality/price ratio. The photodiode is connected to a transimpedance amplifier (TIA), followed by a low-pass filter and a second stage amplifier, as shown in Figure 6.

The maximum input current of 20 nA in converted to an output voltage of 3.0 V, thus a gain of 146.3 V/µA is implemented in the transimpedance amplifier circuit, based on the precision amplifier AD8691 (using only the first stage does not allow to obtain the desired gain, since a large value of Rf is required). This amplifier (AD8691) has a low bias current (typical of 0.2 pA) suitable for the high impedance of the circuit, and a current noise density of 0.05 pA/√Hz, that will produce a noise output of approximately 20 mV at 100 Hz bandwidth, since the filter between the two stages implements a low pass at 100 Hz. 

### 2.3. Characterization Methods

Absorption spectra were recorded with a setup that includes a 200 W quartz tungsten halogen lamp (Newport 6334NS) as light source; a monochromator (Oriel Intruments Cornerstone 74,000); an optical fiber (Newport Standard Grade FS Fiber Optic) used to guide the light through the sample and into the photodiodes box; and a picoammeter (Keithley 487) to measure the photodiode current. This enabled the selection of the best membranes to carry out the remaining tests.

Subsequently, a luminescent analysis was performed. Therefore, a controlled atmosphere was produced inside a small container, made up of pressurized argon and oxygen. For liquid samples, the container was filled with ultrapure water and the same gases were injected into the water volume. 

To obtain homogeneous media, it was necessary to include a small turbine inside the recipient. 

Luminescence analysis was performed with a lock-in amplifier (Zurich Instruments HF2LI) and its software (LabOne), to extract intensity and lifetime of luminescence. Spectral analysis of luminescence was analyzed with a spectrophotometer—*AvaSoft Avaspec ULS2048 XL-RS-EVO*, to carry out the remaining analysis. We considered the linearity, higher absorption, and greater range as fundamental criteria in selecting the best membrane. For this, the selected membrane should have homogeneous distribution of the indicator [20], without crystallizations or aggregate inside the membrane that results in a non-linear response.

The complete sensor based in a photodiode paired with a 647.1 ∓2 nm ThorLabs bandpass filter was tested to evaluate oxygen concentration, and their results were compared with a commercial sensor (Extech DO210). 

## 3. Results and Discussion

During the manufacture of the membranes, three fabrication parameters were considered: the indicator concentration, the solvent used and the final thickness of the membrane. Two different solvents (toluene and THF) were tested; four different indicator concentrations (2, 4, 6 and 8 mg dissolved in 11 g of PDMS resulting in the concentrations of 181, 363, 545 and 727 ppm) were studied, and membrane thicknesses from 0.1 mm up to 2.5 mm were fabricated.

The membranes were first tested for light absorption in the range of 350–600 nm to find the maximum absorption wavelength. Subsequently, the absorption at this wavelength was measured for the membranes of different thicknesses and fabricated with different solvents (THF and toluene). The luminescence spectrum for the different fabricated membranes was recorded and analyzed. The luminescence intensity and the delay were analyzed using the lock-in amplifier. Finally, two sensors with different membranes were fabricated and tested in a gaseous and a liquid environment.

### 3.1. Membrane Absorption

Figure 7 shows a typical absorption result of the fabricated membranes with different concentrations, similar thickness and the same solvent (namely THF). 

There are two clear absorption peaks: the first, located in the Soret zone, at 385 nm (+/−3 nm) and the second in the Q zone, at 535 nm (+/− 3 nm). The emission intensity is higher when the excitation is carried out at wavelengths where higher absorption of the membrane is observed. UV excitation (385 nm) was selected to promote higher luminescence of the membrane. Additionally, this wavelength is sufficiently far from the luminescence wavelength, as demonstrated further.

Figure 8 shows the relationship between absorption (measured at 385 nm) and membrane thickness with the solvent’s toluene and THF, respectively, using different indicator concentrations.

The light absorption depends on the thickness, concentration and solvent used. The higher the concentration and thickness of the membrane, the higher the absorption. The absorption in membranes prepared with THF is almost linear with the thickness, contrary to the results obtained with toluene. Linearity differences may have their origin in the homogeneity of the indicator concentration and homogeneity in the thickness. Membranes made with THF also have slightly higher absorption values than those made with toluene, for the same thickness and indicator concentration. 

### 3.2. Luminescence Spectrum 

Figure 9 shows the variation of luminescence intensity between different toluene and THF-based membranes (indicator concentration of 363 ppm) over a spectrum between 635 nm and 658 nm for three different oxygen concentrations, 0.6%, 1.3% and 2%, with an excitation at 385 nm.

In both solvents, the spectrum is equivalent, and the highest luminescence was obtained at 645 nm. It was also stated that the emission wavelength peak (645 nm) of the indicator is independent of the excitation wavelength, since similar analyses were realized with excitation at 540 nm. Additionally, in both solvents, luminescence decreases as oxygen concentration increases. However, for the toluene solvent, with thicker membranes, luminescence is about 1.8 times higher than in THF-based membranes. An equivalent spectrum was also obtained in membranes with thicknesses between 0.15 and 1.4 mm, using both solvents, and the maximum intensity, the measurement range and the linearity of the inverse of luminescence relative to oxygen concentration (Equation (1)) were measured and results presented in Table 2.

Although toluene membranes have a higher luminescence intensity, THF membranes have higher linearity and homogeneity (Figure 8) and were therefore chosen for the following tests.

### 3.3. Membrane Luminescence Intensity and Lifetime 

Stern–Volmer equations (Equations (1)–(3)) indicate that the variation of oxygen concentration can be measured using intensity (*I*) or lifetime (*τ*) of luminescence. Using an alternated excitation light, both intensity and lifetime can be, respectively, measured by the module and phase of measured luminescence, using a lock-in amplifier. The graphs of Figure 10 show the results of the module and phase of luminescence as functions of oxygen concentration for the membrane 363 ppm 1.4 mm THF in a gaseous environment. The membrane was excited with the UV LED, modulating the intensity at frequencies between 100 Hz and 10 kHz. The resultant photodiode current from the luminescence was converted to voltage using the previously described electronic circuit for amplification (TIA).

The amplitude of the luminescence signal (R) at the output of the transimpedance amplifier decreases with the increase in oxygen concentration, as would be expected by the Stern–Volmer equation (Equation (1)). It is possible to analyse that the higher the excitation frequency, the higher the value of the modulus, representing higher sensor sensitivity. For the oxygen range between 0.5–1% oxygen, it is possible to observe that the membrane is clearly more sensitive, either in modulus (R) and phase (Θ), and its reaction tends to stabilize horizontally, revealing a lack of sensitivity to higher oxygen levels (>6%). An inversely linear relationship between the delay (*τ*) and the oxygen concentration was also expected, which would be reflected as a phase shift in Figure 10. However, only a small phase shift (that corresponds to a delay) was observed for low oxygen concentration (0.5–1%). This lack of agreement with the theory may be due to the reflection and dispersion of the excitation light in the membrane that also could be reaching the photodiode. The reflection and dispersion are in phase with the excitation, thus altering phase measurement. The reflexion/dispersion also affects the amplitude (R), resulting in an offset voltage measured for high oxygen concentrations.

The use of amplitude, through the intensity measurement method, was selected instead of the luminescence delay. This option is justified by the simplicity and the cost of the electronic circuits, since the lock-in amplifier and phase detection could be avoided. Further tests presented below also demonstrate the small effect of photo degradation in this sensor, due to low ON time of the LED light. It is thus possible to conclude that the analysis of the module enables to obtain higher sensibility and a higher measurement range. Although the use of the phase measurement could attenuate the effects of membrane aging, the following tests are carried out to analyse the intensity (modulus) of the signal, given the higher sensitivity observed.

### 3.4. Sensor Luminescence in Gaseous and Liquid Media 

Figure 11 represents the inverse of the photodiode current, whereas the offset of 4.3 nA was already removed, along with the increase in oxygen concentration in a gaseous media for the membrane with 363 ppm of concentration and 1.4 mm of thickness fabricated with THF. As it can be seen, there is a linear trend towards higher oxygen concentration, with a small curvature in the lower oxygen range. It is possible to conclude that the membrane can measure oxygen levels up to 20%, but for higher values of oxygen concentration, the sensibility reduces, thus determining the end of the scale.

Figure 12 and Figure 13 resume the tests performed in liquid media for the membrane with an indicator concentration of 363 ppm and 1.4 mm of thickness, fabricated with THF. Figure 12 represents the digital values obtained by the microcontroller through its ADC, as a function of dissolved oxygen. 

The voltage values were acquired with the 12-bit ADC of the microcontroller (see details on Section 2.2). The ADC uses 12 bits of resolution and a reference (maximum) voltage of 3 V, so its resolution can be calculated in terms of voltage by Equation (6).
(6)$$R_{Volt}=\frac{3.2}{4096}=0.781\;\mathrm{mV}/\mathrm{bit}\;$$

Considering the maximum (3809) and minimum digital values (1936) in the graph of Figure 13, that correspond, respectively, to 3 mg/L and 0.3 mg/L, the mean sensitivity can be calculated by Equation (7).
(7)$$R_{O2}=\frac{3-0.3}{3809-1936}=1.44\;µg/L$$

However, for lower oxygen concentration, a lower value of *R_O_*_2_ is observed, and for higher oxygen concentration (when the sensor approaches the end of the scale), a higher value of *R_O_*_2_ is observed, respectively, 0.48 µg/L and 13 µg/L in the ranges 0.3–1 mg/L and 2–3 mg/L. Electronic noise and external interferences limit the maximum range of the sensor, since luminescence decreases with a higher concentration of oxygen.

Figure 13 represents the variation of the DO values measured by the developed sensor and by the commercial sensor. The sensor response is compensated digitally to achieve a correct output in mg/L. The dashed line represents the ideal theoretical results. There is an almost inverse linear tend relationship between the current/voltage and the concentration of dissolved oxygen, as depicted by the Stern–Volmer relationship. From 3.5 mg/L it is possible to detect noise resulting from the approach to the end of the scale (where the current in the photodiode is reduced). 

An accelerated test of photo degradation was also performed. The selected membrane was submersed in sweater, and illuminated for 7 h, with the selected UV-385 nm LED with a current of 20 mA and a duty cycle of 20% (200 µs ON and 800 µs OFF) corresponding to 1.4 h of continuous illumination. After this test, the membrane showed a reduction of 2% of luminescence when compared with the luminescence before the degradation test.

Since a measurement of oxygen requires a light pulse of the UV LED with a period below 1 ms, these stability results announce that for a degradation up to 2%, more than 5 million measurements can be performed, thus allowing a reasonable lifetime of the membrane, even if a 1-min sampling time of the measurements is required.

## 4. Conclusions

Luminescent PDMS membranes (sensitive to the presence of oxygen) and controlling/acquiring electronics were developed to achieve an optical DO sensor to be used in a marine environment, with a measuring range between 0 and 5 mg/L in liquid medium. It was possible to manufacture oxygen-sensitive membranes from simple fabrication procedures, compatible with standard processes of microfluidics, without the need for complex laboratory equipment.

The developed membranes were based on PDMS doped with the PtOEP indicator, which present a luminescent behavior as described by the Stern–Volmer equation through the luminescence intensity ratio method. A maximum light absorption was verified at 385 nm, and a maximum luminescence at 640 nm, as found in the literature.

It was possible to observe differences between the solvents used in the fabrication processes, concluding that the use of THF can result in membranes with a more linear response to oxygen than toluene. The influence of indicator concentration (181 ppm to 727 ppm) and membrane thickness (0.1 mm to 1.9 mm) was reported, with best results (higher luminescence, higher linearity and higher range) found in thick membranes, with moderate doping. The membrane with 363 ppm of indicator and 1.4 mm thickness, using THF solvent, was selected for further tests.

The sensor was tested in liquid medium and in a gaseous environment, achieving a O_2_ measurement range of 0–5.5 mg/L and 0–21%, respectively. However, due to noise in electronics above the limit of detection in luminescence intensity, only the range of 0–3.3 mg/L in liquid medium and 0.5–20% in gaseous medium can be usable. The sensitivity (with a common 12 bits microcontroller ADC) varies from 0.48 µg/L to 13 µg/L, respectively, in the ranges 0.3–1 mg/L and 2–3 mg/L.

Since luminescence increases with lower oxygen concentration, the limit of detection of the fabricated PDMS membrane is clearly higher than the limit of detection of the electrode-based commercial sensor (Extech DO210), concluding that the PDMS luminescence intensity measurement method is clearly advantageous to oxygen electrode-based sensors for applications where high sensitivity is required at low oxygen levels (below 3 mg/L).

## Figures and Tables

**Figure 1 sensors-21-05645-f001:**
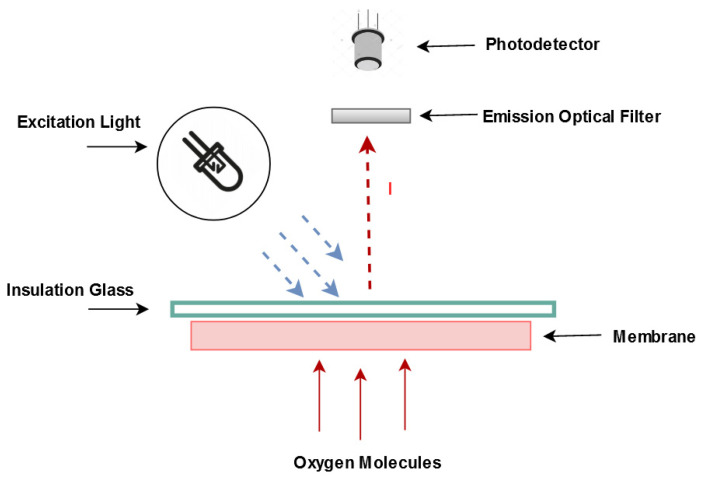
Diagram of a typical dissolved oxygen optical sensor.

**Figure 2 sensors-21-05645-f002:**
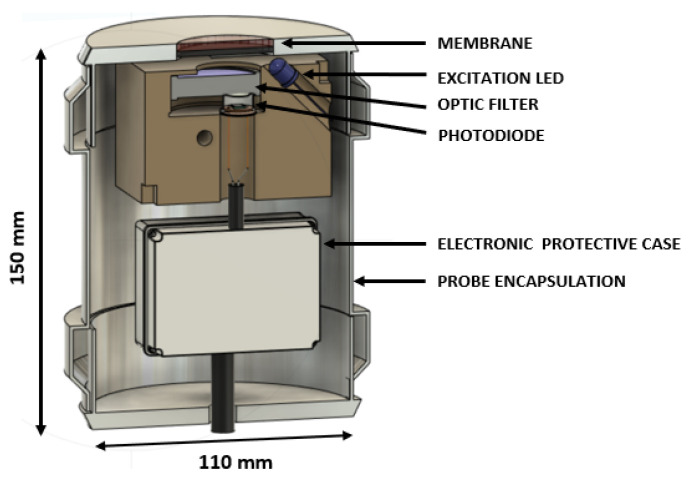
Schematic representation of the main components and encapsulation of the oxygen sensor.

**Figure 3 sensors-21-05645-f003:**
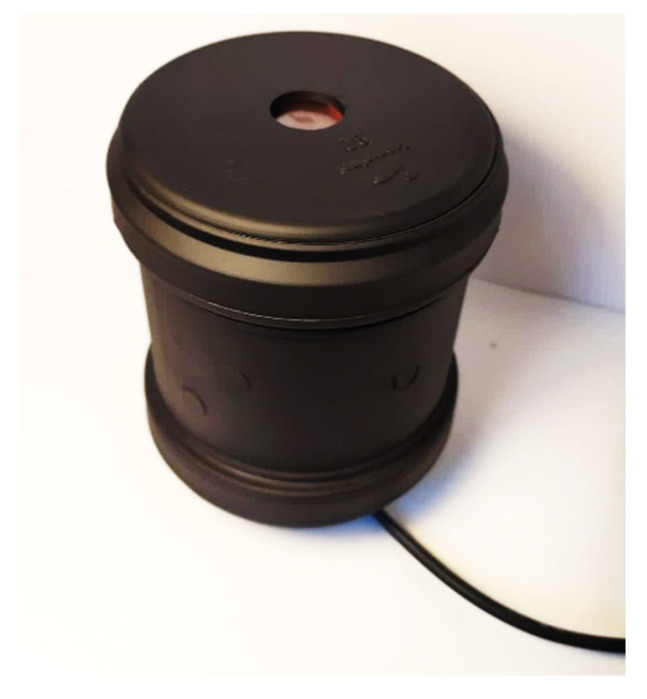
Photograph of the fabricated device.

**Figure 4 sensors-21-05645-f004:**
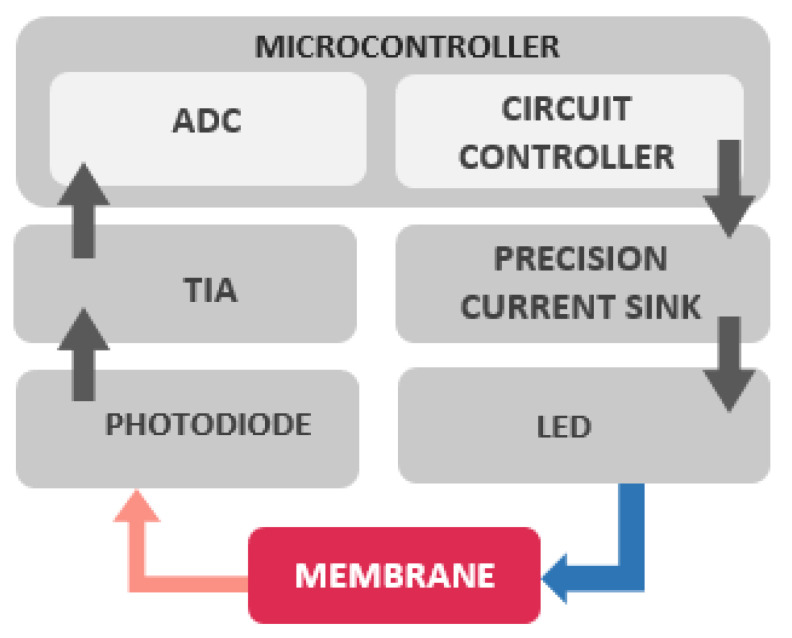
Brief description of hardware blocks.

**Figure 5 sensors-21-05645-f005:**
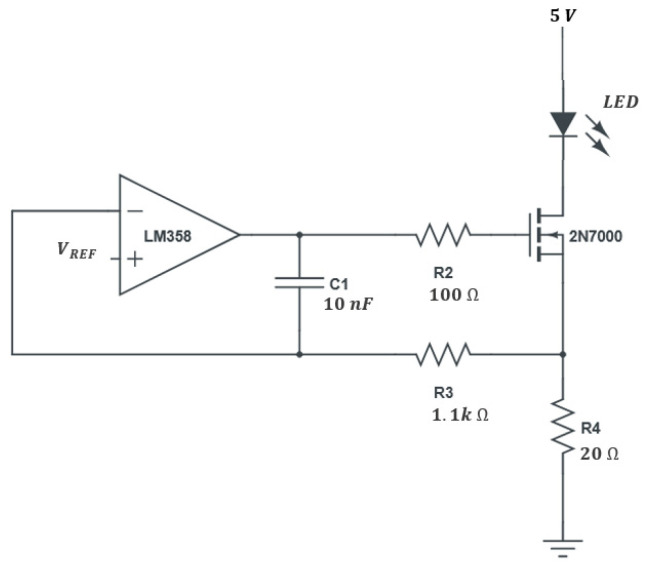
Constant current sink used to feed the excitation LED.

**Figure 6 sensors-21-05645-f006:**
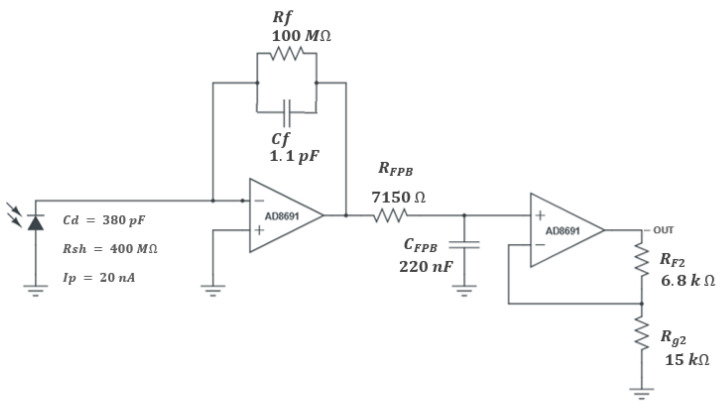
The signal condition circuit, composed of the transimpedance amplifier, a low-pass first order filter, and a second amplifier stage.

**Figure 7 sensors-21-05645-f007:**
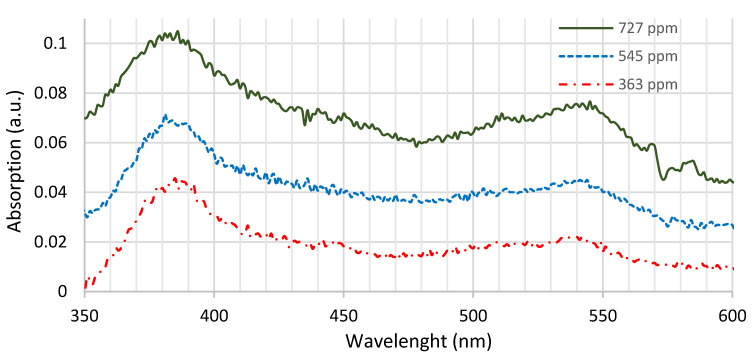
Typical absorption of PtOEP for different concentrations with similar thicknesses, namely 0.2 mm for the membranes of 363 ppm and 545 ppm of concentration and 0.15 mm for the membrane of 727 ppm of concentration.

**Figure 8 sensors-21-05645-f008:**
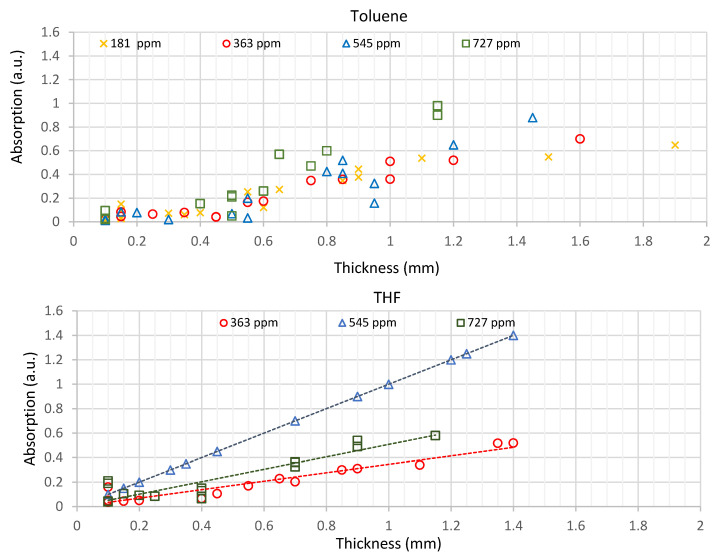
Relationship between absorption and thickness, for membranes manufactured using the solvent toluene and THF, with indicator concentrations of from 181 ppm to 727 ppm.

**Figure 9 sensors-21-05645-f009:**
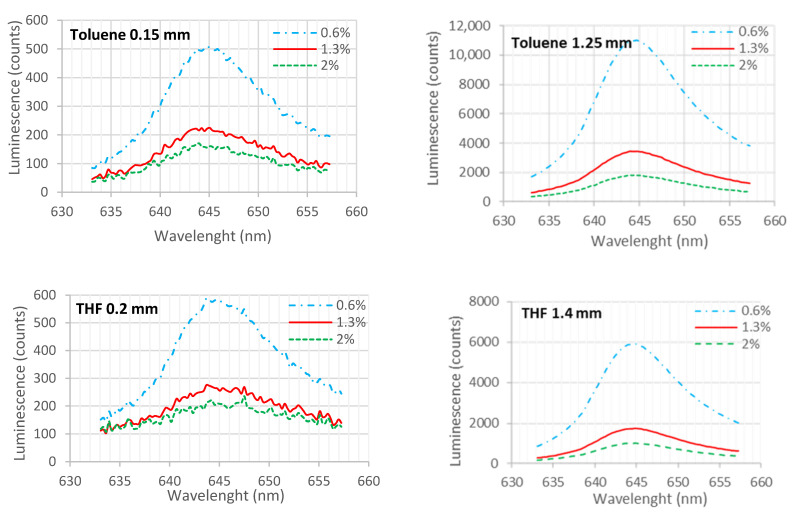
Luminescence spectrum of THF and toluene-based membranes with a concentration of 363 ppm, for thin and thick membranes, respectively, 0.15–0.20 mm and 1.24–1.4 mm. Each line in the graphs was obtained with different oxygen concentration (gaseous media) with oxygen concentration of 0.6%, 1.3% and 2%.

**Figure 10 sensors-21-05645-f010:**
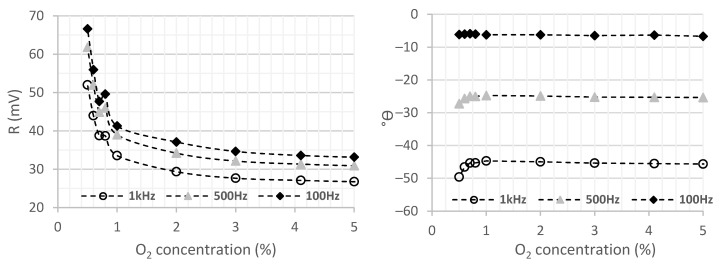
Phase shift of the sensor signal as a function of oxygen variation, for the 363 ppm concentration membrane with a thickness of 1.4 mm and THF in a gaseous environment.

**Figure 11 sensors-21-05645-f011:**
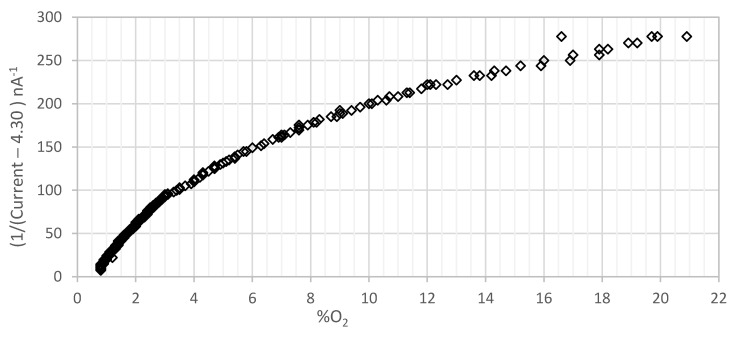
Inverse of the current intensity along the increase in oxygen concentration, in a gaseous environment, for the 363 ppm membrane with a thickness of 1.4 mm and THF. The 4.3 nA offset has been removed, and the inverse of the photodiode current is shown.

**Figure 12 sensors-21-05645-f012:**
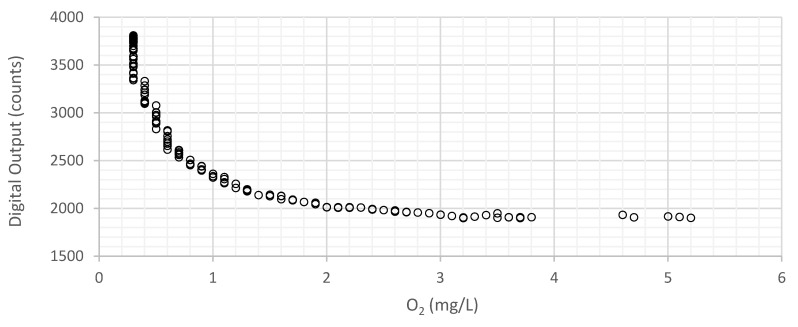
ADC values as a function of oxygen concentration for the membrane with 363 ppm of indicator with a thickness of 1.4 mm, manufactured with THF, in a liquid environment.

**Figure 13 sensors-21-05645-f013:**
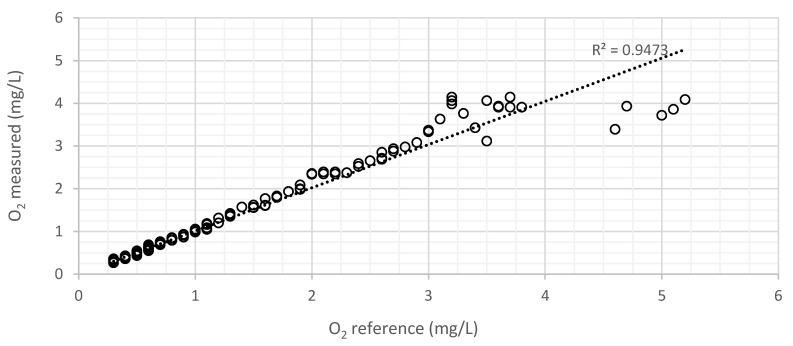
Variation of the DO values measured by the developed sensor as a function of the reference DO values (measured with the commercial sensor) for the membrane with a concentration of 363 ppm with 1.4 mm and manufactured with THF, in a liquid environment, obtained with the Stern–Volmer equation (Equation (1)).

**Table 1 sensors-21-05645-t001:** Polymer proprieties (permeability, diffusion, and solubility).

Polymer	$P\left(\times 10^{13}\right)$ $cm^{3}\;\left(STP\right)\;cm^{-2}s\;cm\;Hg$	$D\left(\times 10^{6}\right)\;cm^{2}s^{-1}$	$S\left(\times 10^{6}\right)$ $cm^{3}\;\left(STP\right)\;cm^{-3}cm\;Hg^{-1}$	Ref.
PDMS Poly(dimethylsiloxane)	695	40	24	[18]
PMSP Poly(1-trimethylsily-1-propyne)	7700	47	170	[17]
PS Polystyrene	2.63	--	--	[19]
PMMA Poly(methylmethacrylate)	9	10	8.5	[18]
PVC Poly(vinyl chloride)	0.34	1.2	2.9	[18]
*PiBMA* Poly(isobutylmethacrylate)	20	--	--	[18]
Poly(2,2,2-trifluoroethylmethacrylate)	32	15	0.27	[16]
EC Ethylcellulose	11	0.639	1.73	[18]
CAB Cellulose acetibutyrate	3.56	--	--	[18]
CA Cellulose acetate	5.85	--	--	[18]

**Table 2 sensors-21-05645-t002:** Characterization of membranes.

Solvent	Indicator Concentration (ppm)	Thickness (mm)	Maximum Intensity (Counts)	Linearity	Range (%O_2_)
Toluene	363	0.15	566	Linear	0.4–3%
		1.25	10,084	Non-linear	0.4–15%
	545	0.15	282	Linear	0.4–4.5%
		1.2	6008	Linear	0.4–14%
	727	0.15	4444	Linear	0.4–4%
		1.15	6641	Non-linear	0.4–10%
THF	363	0.2	787	Non-linear	0.4–3%
		1.4	4689	Aprox. linear	0.4–21%
	545	0.2	715	Non-linear	0.4–4%
		1.4	4414	Non-linear	0.4–12%
	727	0.15	576	Non-linear	0.4–2.9%
		1.15	760	Non-linear	0.4–18%

## Data Availability

Not applicable.

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
