# Peer review of "PtOEP–PDMS-Based Optical Oxygen Sensor"

_sensors, 2021, doi:10.3390/s21165645_

Round 1
Reviewer 1 Report
In this work, the author prepared the luminescent PDMS membranes to achieve an optical DO sensor for marine environment application. The measuring range in liquid medium was confirmed to between 0 to 5 mg/L with a simple fabrication procedure. The authors also investigated the differences of the products using various solvents for the fabrication and the THF was found to provide the more linear response when compared with the toluene based process. The conclusion can be well supported by the experimental results. My comments are as below.
- The novelty of this work is not clear. Even the systematical studies on the performance have been carried out, it seems that some of the experimental results are well-known in the field. The significance, e.g. in the introduction, of the work should be re-considered and delivered in a better manner.
- The abstract should be reformatted. For example, the excitation wavelength and bandpass photodiode are not necessary to appear in the abstract.
- The authors claim that ‘the selected membrane would be the most homogeneous, and its luminescent properties would be optimal for marine application’. Is there any direct results that can support such conclusion about the homogeneity and the optimal for marine application?
- As mentioned, the higher concentration of the indicator would lead to a better light absorption. If so, what will happen if the indicator concentration was further increased to above 727 ppm?
- How about the performance of the device when it is immersed in seawater, instead of the ultrapure water? This should be important to convince the potential of use in marine environments.
- All figure citation should appear in the context. Please check the citation of Figure 8 and Figure 9, and else if any.
- There are lots of ‘Error! Reference source not found’ in the manuscript. Please double check.
- For the axis in the plots, e.g. Figure 10, the intervals of the number should be smaller so that the information from the axis can be more precise. Please check across the manuscript.
- In Figure 13, how about the linearity (R square) of the data?
Author Response
Author's reply in the attached file.

Reviewer 2 Report
In this manuscript the authors present an optical O2 sensor based on PtOEP in PDMS.
Overall this manuscript is of very low significance and barely adds any new insights to a field that is very well established. Optical O2 sensors are commercially available, and the commercial sensors outperform the presented sensor in multiple ways. The authors claim in the abstract that biofouling can be improved using this sensor, but fail to show any relevant data. This would have been the only “novel” addition to the state of the art, but as mentioned this is not addressed in the manuscript.
I recommend rejection of the manuscript. A detailed list of comments can be found below.
- Introduction reads like a mini review. What is the aim? The educated reader knows most of the presented information. It remains unclear where the niche of this sensor should be.
- OMZs are well studied using both optical (e.g. LUMOS sensors) as well as electrochemical (STOX sensors). The presented sensor does not appear suited to measure trace concentrations anyway. Why use OMZs as an example?
- It remains unclear if the sensor membrane (called optode by the way) or the sensor module is the novelty.
- In experimental section several things are unclear like the solvent removal. Why? And how was that controlled? The mentioned LED has a maximum at 385nm and not 350nm as stated in the manuscript. Just to name a few mistakes and things that are unclear.
- Absorption is by definition unitless. Why is it presented in counts? Why is the baseline increasing with concentration?
- The stability of the dye in the polymer is questionable. It is well known that the hydrophilic PDMS is not a great matrix for PtOEP. A long term stability test is missing.
- Figures are mixed up in the text and often it just says “Error! Reference source not found”. This indicates that the authors did not check the final version prior to submission.
- Why is lifetime measurement not possible? This is standard and promoting a device that is not capable is not good. Stating that reflected light is an issue is not an excuse. If that is the case filters should be checked etc.
Author Response
Author's reply in the attached file Reviewer_2.docx

Round 2
Reviewer 1 Report
The authors have appropriately revised the manuscript according to the raised comments. Some typos should be further checked, e.g. ' These option is justified' (These options are justified) and 'Further tests presented bellow' (Further tests presented below) in page 11 .
Author Response
Dear Reviewer,
We fully appreciate the positive comments to our paper. We have read carefully the manuscript and corrected the typos. Thanks again for your valuable revision.
Best regards,
Graça Minas
Reviewer 2 Report
The authors barely changed the manuscript. In essence all my previous issues remain.
Changing the focus from OMZ to microfluidics is useless and demonstrates that the authors have no clear idea on how to use this sensor. Also again the proposed readout unit is totally unsuited for microfluidic readout. The readout unit may be able to measure in bulk, but not in small volumes.
All in all this manuscript has not improved and still should not be published.
Author Response
Dear reviewer,
First we would like to thank you for taking your valuable time reading our manuscript.
Previously, we addressed your major manuscript revision, or justified our position, when applicable. Also, some new work was also added (preliminary photo degradation tests).
In the current reply we just want to clarify our position regarding your negative comments, since details about a possible revision were not provided.
We disagree with your comment “Changing the focus from OMZ to microfluidics is useless and demonstrates that the authors have no clear idea on how to use this sensor”. Our main focus is the same, “fabrication and characterization of oxygen sensitive membranes based on PtOEP/PDMS and their readout and control electronics, to design low cost, low energy consumption, easy fabrication oxygen sensors”. The focus on compatibility of fabrication process with microfluidics was an added value on previous revision, in order to get broader audience and improve innovation, but as you notice the work is the same.
Our previous application in Oxygen Minimum Zones (OMZ), resulted from the expected low limit of detection of your sensor. As your suggestion, removed OMZ specific application, since the detection limit of sensor in the range below 0.3 mg/L was not fully practical demonstrated, only an obvious (in our point view) extrapolation of current results. The fully demonstration would require us the acquisition of new equipment, not possible in a fast publishing process as required by Sensors journal.
We totally agree that “the proposed readout unit is totally unsuited for microfluidic readout and may be able to measure in bulk, but not in small volumes”. However, in our opinion and experience in microsystems, the fabrication of oxygen sensitive membranes in the same material and technology of microfluidics devices is the innovation in our work, redesign it to a lab-on-chip device would be a minor task, and scale down the electronics to the required dimension doesn’t even worth publication. The significance of dissolved oxygen sensors incorporated in microfluidic scales can be seen at [1, 2] below. However, we accept that opinion from researchers in other research areas could be different.
And we also agree your previous comment, that some optodes are available commercially (example the well know LUMOS), some of them with better characteristics. However, most publications about commercial available optodes describe the integration of commercial membranes, or the electronics are not fully described, and as result, work cannot be fully replicated, since details are not presented as in our work.
[1] Bunge, Frank, et al. "Microfluidic oxygen sensor system as a tool to monitor the metabolism of mammalian cells." Sensors and Actuators B: Chemical 289 (2019): 24-31. https://doi.org/10.1016/j.snb.2019.03.041
[2] Mousavi Shaegh, Seyed Ali et al. “A microfluidic optical platform for real-time monitoring of pH and oxygen in microfluidic bioreactors and organ-on-chip devices.” Biomicrofluidics vol. 10,4 044111. 26 Aug. 2016, doi:10.1063/1.4955155